# Overexpression of miR-375 and L-type Amino Acid Transporter 1 in Pheochromocytoma and Their Molecular and Functional Implications

**DOI:** 10.3390/ijms23052413

**Published:** 2022-02-22

**Authors:** Jacopo Manso, Loris Bertazza, Susi Barollo, Alberto Mondin, Simona Censi, Sofia Carducci, Alfonso Massimiliano Ferrara, Isabella Merante Boschin, Stefania Zovato, Francesca Schiavi, Michele Gregianin, Gianmaria Pennelli, Maurizio Iacobone, Caterina Mian

**Affiliations:** 1Endocrinology Unit, Department of Medicine (DIMED), Padua University, 35121 Padua, Italy; loris.bertazza@unipd.it (L.B.); susi.barollo@unipd.it (S.B.); alberto.mondin.1@phd.unipd.it (A.M.); ssimonacensi@gmail.com (S.C.); sophie.carducci@gmail.com (S.C.); isabella.meranteboschin@unipd.it (I.M.B.); 2Familial Cancer Clinic and Oncoendocrinology, Veneto Institute of Oncology IOV-IRCCS, 35128 Padua, Italy; massimiliano.ferrara@iov.veneto.it (A.M.F.); stefania.zovato@iov.veneto.it (S.Z.); francesca.schiavi@iov.veneto.it (F.S.); 3Nuclear Medicine Unit, Veneto Institute of Oncology IOV-IRCCS, 31100 Treviso, Italy; michele.gregianin@aulss2.veneto.it; 4Surgical Pathology and Cytopathology Unit, Department of Medicine (DIMED), Padua University, 35121 Padua, Italy; gianmaria.pennelli@unipd.it; 5Department of Surgical, Oncological and Gastroenterological Sciences (DiSCOG), Padua University, 35121 Padua, Italy; maurizio.iacobone@unipd.it

**Keywords:** pheochromocytoma, microRNA, miR-375, L-type amino acid transporter, 18F-dihydroxyphenylalanine

## Abstract

Pheochromocytoma (Pheo) is a tumor derived from chromaffin cells. It can be studied using 18F-dihydroxyphenylalanine (DOPA)—positron emission tomography (PET) due to its overexpression of L-type amino acid transporters (LAT1 and LAT2). The oncogenic pathways involved are still poorly understood. This study examined the relationship between ^18^F-DOPA-PET uptake and LAT1 expression, and we explored the role of miR-375 and putative target genes. A consecutive series of 58 Pheo patients were retrospectively analyzed, performing ^18^F-DOPA-PET in 32/58 patients. Real-time quantitative PCR was used to assess the expression of LAT1, LAT2, phenylethanolamine N-methyltransferase (PNMT), miR-375, and the major components of the Hippo and Wingless/Integrated pathways. Principal germline mutations associated with hereditary Pheo were also studied. Pheo tissues had significantly higher LAT1, LAT2, and PNMT mRNA levels than normal adrenal tissues. MiR-375 was strongly overexpressed. Yes-associated protein 1 and tankyrase 1 were upregulated, while beta-catenin, axin2, monocarboxylate transporter 8, and Frizzled 8 were downregulated. A positive relationship was found between ^18^F-DOPA-PET SUV mean and LAT1 gene expression and for 24 h-urinary norepinephrine and LAT1. This is the first experimental evidence of ^18^F-DOPA uptake correlating with LAT1 overexpression. We also demonstrated miR-375 overexpression and downregulated (Wnt) signaling and identified the Hippo pathway as a new potentially oncogenic feature of Pheo.

## 1. Introduction

Pheochromocytoma (Pheo) is a tumor derived from adrenomedullary chromaffin cells, originating in adrenal medulla or extra-adrenal sites (paraganglioma). It is typically a functioning catecholamine-secreting tumor that produces epinephrine, norepinephrine, and/or dopamine [1]. These tumors may very rarely be biochemically silent. At least one in three cases of Pheo are familial. Only about one in ten are malignant [2].

The diagnosis of Pheo usually relies on biochemical evidence of an excessive release of catecholamines, together with imaging findings of the tumor. Plasma or 24-h urinary metanephrine and normetanephrine assays are the most reliable markers of Pheo, but medication or diet can give rise to false-positive test results [1,3,4]. The latest Endocrine Society guidelines on Pheo recommend computed tomography (CT) rather than magnetic resonance imaging (MRI) [1]. ^123^I-metaiodobenzylguanidine (MIBG) scintigraphy, ^18^F-dihydroxyphenylalanine (DOPA)—positron emission tomography (PET), ^68^Ga-1,4,7,10-tetra-azacyclododecane-1,4,7,10-tetraacetic acid-octreotate (DOTA-TATE)—PET, and ^18^F-fluorodeoxyglucose (FDG)—PET are also useful functional imaging tools for confirming a diagnosis of Pheo and for detecting any metastatic disease [1,5,6,7]. In this scenario, the rationale for using the ^18^F-DOPA tracer is based on the fact that neuroendocrine tumors are capable of taking up, decarboxylating, and storing amino acids and their biogenic amines such as L-DOPA [8]. L-DOPA uptake in the cytoplasm of neuroendocrine cells is done by an L-type amino acid transporter (LAT) system [7]. For imaging purposes, L-DOPA can be radiolabeled with an ^18^F positron emitter isotope in the sixth position to form ^18^F-DOPA, which moves in the same catecholamine metabolism pathway as its natural counterpart, mirroring its endogenous kinetics [9]. There are two main LAT isoforms belonging to the SLC7 transporter gene family, i.e., LAT1 (or SLC7A5) and LAT2 (or SLC7A8). An increased ^18^F-DOPA uptake and tracer retention by Pheo is due to the tumor’s LAT1 and LAT2 overexpression, as previously shown by our group [10].

^18^F-DOPA-PET could be useful for confirming a diagnosis of Pheo, its initial staging, and during patient follow-up. Fiebrich et al. showed that ^18^F-DOPA-PET is better than ^123^I- MIBG (93% vs. 76%, *p* < 0.001) in identifying Pheo and found a positive correlation between tumor burden on ^18^F-DOPA-PET and the blood normetanephrine level [11].

Epigenetic changes are defined as stably inherited modulations in gene expression without any modification of the DNA sequence [12]. DNA methylation and microRNA (miRNA) regulation are the best known epigenetic changes. MiRNAs are small single-stranded non-coding RNAs that take part in regulating biological processes by inhibiting gene expression at a post-transcriptional level, thereby influencing cell differentiation, growth, and death, and they are consequently important in the diagnosis, pathogenesis, and prognosis of cancer, including endocrine tumors [13,14].

It has recently been demonstrated that miRNA dysregulation is implicated in the genesis of Pheo as well. Research conducted so far has focused on the possible role of miRNAs in differentiating ab initio and metastatic Pheo from its benign counterpart. Several research groups compared the miRNA expression signature of benign and malignant Pheo, finding miR-15a and miR-16 underexpressed, and miR-483-5p, miR-183, and miR-101 overexpressed in malignant Pheo [15,16] Another issue investigated concerns the possible role of miRNAs in distinguishing between sporadic and familial syndromic Pheo. Tombol et al. found that miR-139-3p, miR-541, and miR-765 could distinguish von Hippel-Lindau (VHL)-related Pheo from sporadic Pheo, while miR-885-5p and miR-1225-3p were overexpressed in multiple endocrine neoplasia type 2 [17]. In a more recent paper, De Cubas et al. identified miR-133b as specific to VHL, miR-488 and miR-885-5p as related to multiple endocrine neoplasia type 2, and miR-183 and miR-96 as specific to succinate dehydrogenase complex B (SDHB) [18].

MiR-375 dysregulation is emerging as a novel epigenetic feature involved in neuroendocrine tumor differentiation. Hudson et al. reported miR-375 overexpression in medullary thyroid cancer (MTC) tissues, associated with a downregulated growth inhibitor yes-associated protein 1 (YAP1) and monocarboxylate transporter 8 (MCT8, a thyroid hormone transporter), which were identified as potentially important downstream target genes of miR-375 [19]. Our group confirmed miR-375 overexpression (at both tissue and serum levels) and its relationship with YAP1 nuclear loss in a larger cohort of MTC patients, supporting the impression that miR-375 represents a new molecular marker of neuroendocrine tumorigenesis. Remarkably, no overlap was found between miR-375 overexpression in cancer samples and normal thyroid tissues and sera, which would point to a potential diagnostic role for this miRNA [20,21]. On the other hand, He et al. found miR-375 downregulated by comparison with normal adrenal cortex in adrenal aldosterone-producing adenomas [22].

The aim of the present study was to confirm our previous findings regarding LAT1 and LAT2 expression in Pheo in a larger series of patients and to examine the relationship between LATs and the tumor secretory properties and ^18^F-DOPA-PET uptake parameters.

Given that the role that miR-375 seems to have in the genesis of neuroendocrine tumors and the lack of data on its role in Pheo, we also investigated its expression in Pheo and its putative downstream targets.

## 2. Results

The baseline characteristics of the 58 patients with Pheo included in our study are summarized in Table 1. The median tumor size was 40 mm. There were three patients with malignant Pheo and only two cancer-related deaths. The median 24-h urinary normetanephrine/metanephrine ratio was 2, meaning that there was a prevalence of norepinephrine-secreting Pheo in our sample. A germline mutation was identified in about one in three of our patients.

### 2.1. L-Type Amino Acid Transporter Expression, Secretory Profile and ^18^F-DOPA-PET

Pheo specimens showed a significant increase in LAT1 (*p* < 0.0001), LAT2 (*p* = 0.0023), and phenylethanolamine N-methyltransferase (PNMT) (*p* < 0.0001) mRNA levels, by comparison with normal adrenal tissues (Figure 1A–C). The median upregulation compared with their normal counterparts was 11-fold for LAT1, 2.6-fold for LAT2, and 34.7-fold for PNMT.

Positive correlations emerged between: 24-h urinary norepinephrine and LAT1 expression (rho = 0.384, *p* = 0.0158); 24-h urinary metanephrine and PNMT gene expression (rho = 0.680, *p* < 0.0001); metanephrine expression and tumor size (rho = 0.327, *p* = 0.0170); and 24-h urinary normetanephrine expression and tumor size (rho = 0.564, *p* < 0.0001) (Figure 2A–D). A statistically significant negative relationship was found between 24-h urinary normetanephrine and PNMT gene expression (rho = −0.286, *p* = 0.0352) (Figure 2E).

We also found a positive relationship between the ^18^F-DOPA-PET SUV mean value and LAT1 gene expression (rho = 0.426, *p* = 0.0321) (Figure 3), while there were no associations with the SUV max. value.

### 2.2. MiR-375 and Downstream Gene Targets

We found miR-375 clearly upregulated in Pheo compared with normal adrenal tissues (*p* < 0.0001), with a median 55-fold overexpression (Figure 4). When we examined the gene expression of possible downstream targets of miR-375 (Figure 5A–E), we found that pathological tissues overexpressed YAP1 (*p* < 0.0001, median 25-fold overexpression), and tankyrase 1 (TNKS1) (*p* < 0.0001, median 3-fold overexpression). On the other hand, beta-catenin was downregulated (*p* = 0.0181, mean 1.3-fold underexpression), and so were axin2 (*p* < 0.0001, median 2.2-fold underexpression) and MCT8 (*p* < 0.0001, median 3.3-fold underexpression) compared with normal expression levels. There was a strongly negative correlation between the miR-375 expression levels and Frizzled 8 (FZD8) (rho = −0.485, *p* = 0.0001) in Pheo specimens (Figure 5F). No statistically significant differences between sporadic and familial cases came to light in the expression levels of all mRNAs and miRNAs investigated.

## 3. Discussion

Pheo is typically a catecholamine-secreting tumor, and at least one in three cases are hereditary, with an underlying germline mutation. The proportions of familial and sporadic forms of Pheo in our study sample are consistent with the literature, but there were slightly more benign cases than usually reported (95% vs. 90% in literature), and we found a prevalent norepinephrine production pattern, as confirmed by the 24-h urinary normetanephrine/metanephrine ratio of 2.

Our previous work produced the first experimental evidence of LAT transporter overexpression in Pheo, especially for the LAT1 isoform. This provides the molecular basis for the increased DOPA uptake seen in Pheo tumor cells, although we did not draw any comparisons with functional imaging data [10]. In the present study on a larger sample, we first confirmed that LAT1 and LAT2 are overexpressed in Pheo by comparison with normal adrenal medulla, and this explains the Pheo cells’ higher ^18^F-DOPA uptake. Then, for the first time to our knowledge, we showed a positive correlation between LAT1 expression levels and ^18^F-DOPA-PET uptake by the SUV mean value, thereby validating the widely accepted hypothesis that ^18^F-DOPA enters neuroendocrine cells via the LAT system.

A previous paper focused on the relationship between LAT1/Cd98hc expression and ^18^F-DOPA uptake, demonstrating a conflicting association between LAT1 protein expression and ^18^F-DOPA-PET uptake in a mixed population of Pheo and paraganglioma cases, carrying SDHx mutations, speculating a possible role of other LAT family transporters, such as LAT4 [23]. Our population included only Pheo with just four samples carrying SDHx mutations: this could justify the consistently strong positive correlation between LAT1 and the ^18^F-DOPA-PET SUV mean. In such a context, LAT1 overexpression could be seen as a marker of biological differentiation in Pheo.

We also found a positive correlation between 24-h urinary norepinephrine and LAT1 expression levels. This statistically significant link between LAT1 overexpression and high levels of urinary norepinephrine metabolites might point to this amino acid transporter being an important molecular step not only in these tumors’ greater ^18^F-DOPA uptake but also (to some degree, at least) in their increased catecholamine secretion. In fact, we found no such positive correlation between LAT1 overexpression and epinephrine metabolites. Instead, we found an upregulation of PNMT—an enzyme found primarily in the adrenal medulla converting norepinephrine to epinephrine and fundamental to epinephrine production from a functional standpoint [24]. This was to be expected, given the pathophysiology of Pheo, and is consistent with other reports [25]. Looking at the secretory profile of our Pheo tissues, the positive correlation between PNMT expression and 24-h urinary metanephrine and the negative correlation between PNMT expression and 24-h urinary normetanephrine both make sense.

As in the case of other hormone-secreting tumors, urinary excretion of metanephrines and normetanephrine correlated positively with tumor size in our sample of patients with Pheo, as reported elsewhere [26].

MiR-375 is emerging as a new epigenetic alteration involved in neuroendocrine tumorigenesis. An overexpression of miR-375 in MTC was demonstrated both by Hudson et al., and by our own group [19,20]. YAP1 and MCT8 were found downregulated in this neuroendocrine thyroid tumor, leading them to be identified as potential target genes of miR-375. Many other miRNAs possibly implicated in the genesis of Pheo have already been studied [15,16,17,27], but miR-375 was only explored in one paper by Fishbein et al. and only in a subset of patients with a rare form of Pheo harboring a MAML3 fusion gene mutation [28]. In this particular population, the authors identified an underexpression of miR-375 with an overexpression of Wingless/Integrated (Wnt) signaling. This prompted us to extend the analysis of miR-375 in a series of more generic Pheo, in which miR-375 was also clearly overexpressed, as seen in MTC [19,20]. It emerged that miR-375 was unable to distinguish between malignant and benign forms or between hereditary and sporadic Pheo in our series. MiR-375 is a validated negative regulator of the Wnt pathway, targeting beta-catenin and FZD8 [29]. FZD8, a member of the Frizzled receptor family, might activate canonical or non-canonical Wnt signaling [30], and dysregulation of the Wnt signaling pathway and/or of Hippo pathway components has been described in several types of cancer [31,32]. We consequently analyzed the gene expression levels of the major components of these pathways in Pheo specimens to shed more light on the possible putative target of miR-375 in this tumor.

Unlike other neuroendocrine tumors (such as MCT), what emerged from our data is that the Hippo pathway may have an oncogenic role in Pheo, judging from the overexpression of YAP1 and TNKS1. YAP1 is a transcriptional coactivator capable of both oncogenic and tumor-suppressive activity depending on the tissue context [33]. For instance, Tu et al. showed that YAP1 hyperactivation was a major driver of the squamous subtype of pancreatic ductal adenocarcinoma [32]. Angiomotins (AMOTs) are YAP downregulators, and recent studies showed that inhibiting TNKS1 hampers YAP oncogenic signaling by stabilizing AMOTs through an inhibitory action on their tankyrase RNF146 axis-mediated degradation [34,35,36]. This means that, even if YAP1 is considered a downstream target gene of miR-375, we could speculate—in line with our findings—that YAP1 signaling in Pheo is hyperactivated by TNKS1 via other, as yet unknown tumor-specific molecular pathways. Further studies are needed to clarify this point.

On the other hand, some components of the Wnt pathway (beta-catenin and axin2) were always downregulated in our Pheo patients, meaning that Wnt signaling may only be significant for tumorigenesis in cases of MAML3-mutated Pheo. Since various Wnt pathway effectors are well-recognized targets of miR-375, their downregulation in Pheo might well be a consequence of miR-375 overexpression. This hypothesis would find strong support in the robust negative correlation that we found between miR-375 and FZD8.

Another putative downstream target of miR-375 is MCT8 [19]. In fact, we found MCT8 underexpressed in Pheo specimens. There is a growing body of evidence to indicate that thyroid hormones can act as oncosuppressors in many tumors (of the breast, colon or thyroid, for instance) through their thyroid nuclear receptors (TRs) [37]. Thyroid hormones dampen a tumor’s accelerated cell proliferation by inhibiting activating protein 1 [38,39], and various cancers have revealed inactivating mutations in TRs that reduce the action of thyroid hormones. When wild-type TR expression was restored in liver and breast cancer cell lines, there were signs of a slowing of the tumors’ growth and a suppression of their invasiveness and metastatic potential [40].

In conclusion, the present study confirms in a larger population that LAT1 and LAT2 are upregulated in Pheo. It provides the first experimental proof of a quantitative correlation between ^18^F-DOPA uptake and LAT1 expression levels and the first evidence of miR-375 overexpression in Pheo, with a consensual downregulation of the Wnt signaling. The present report also paves the way to studies on the Hippo pathway as a possible new oncogenic driver in Pheo.

We are aware that our study has some limitations: particularly, its retrospective nature and the molecular findings related to patients with a low rate of malignant cases. We hope that in the future the study of the Hippo pathway could lead to the development of new target approaches for the treatment of Pheo. Finally, circulating miRNAs emerged as novel biomarkers linked to cancer diagnosis and prognosis. Indeed, circulating miR-375 levels were significantly higher in plasma samples of MTC patients in comparison with those from healthy subjects and were able to distinguish between patients in remission and those with recurrent or persistent MTC [41]. Nevertheless, to date, miRNAs do not have sufficient validated data to be used as a diagnostic or prognostic tool in the everyday clinical practice. It would be interesting to evaluate in further studies the possible diagnostic or prognostic role of circulating miR-375 in larger series of Pheo patients, including a suitable number of malignant cases.

That said, our findings stem from the analysis of a small series of patients and therefore need to be validated in larger cohorts.

## 4. Materials and Methods

### 4.1. Patients

A serum and tissue bank has been operating at Padua University Hospital since 2005. All patients undergoing surgery for endocrine diseases are asked to consent to their tissue and serum samples (the latter obtained before surgery) being collected and stored for research purposes. From this bank we retrospectively and consecutively collected 58 patients with Pheo confirmed histopathologically between 2007 and 2018 (21 men and 37 women; median age 53, range 25–83 years). We defined Pheo as benign when there was no evidence of metastasis during at least two years of follow-up after surgery or malignant when there was a documented metastasis to a region where adrenal tissue would not be expected, neither at initial presentation nor during follow-up [15]. Pheo tissue samples were collected at the time of surgery, snap frozen in liquid nitrogen, and then stored at -80 °C. Matched normal adrenal medulla tissue samples were obtained. The median patient follow-up was 45 months with an interquartile range (IQR) of 23–80 months.

The study was conducted in accordance with the guidelines of the Declaration of Helsinki. All data were collected retrospectively, and all tests were performed as part of standard patient care, so this study did not need ethical committee approval. All patients gave their written informed consent to the publication of the data included in this article.

High-performance liquid chromatography (HPLC) assay, coupled with electrochemical detection using a Chromsystems 5000^®^ kit (Chromsystems Instruments & Chemicals, Germany), was used to measure 24-h urinary metanephrine, normetanephrine, epinephrine, and norepinephrine in samples collected for diagnostic purposes prior to surgery. ^18^F-DOPA-PET/CT was performed in 32/58 (55%) patients for initial diagnosis and staging.

### 4.2. ^18^F-DOPA-PET/CT Imaging Protocol

A combined PET/CT scanner was used. Patients fasted for at least 3 h before ^18^F-DOPA was administered intravenously (3 MBq/kg). The ^18^F-FDOPA-PET/CT acquisition protocol encompassed a delayed whole-body acquisition (starting about 60 min after the radiotracer’s injection) from the top of the skull to the upper thigh (3 min/step). Volumetric regions of interest were placed over the areas of ^18^F-DOPA uptake in the Pheo and liver. SUV max. (maximum voxel intensity within the volumetric region) and SUV mean (the average ^18^F-DOPA uptake in the tumor) were calculated.

### 4.3. Germinal Mutation Screening by Direct Sequencing

At a germinal level, all exons of the succinate dehydrogenase complex (*SDHA*, *SDHB*, *SDHC*, *SDHD*), *VHL*, and exons 5, 8, 10, 11, and 13–16 of the *REarranged during Transfection* (RET), *MAX*, *TMEM127*, and *neurofibromatosis type I* (NF-1) were examined by direct sequencing with the Sanger method.

### 4.4. RNA Extraction

Total RNA, including small RNAs (17–200 nt), were isolated from fresh snap-frozen samples using the DirectZol RNA Miniprep Plus Kit (Zymo research, Irvine, CA, USA) according to the manufacturer’s instructions.

### 4.5. miRNA Quantitative Real-Time Polymerase Chain Reaction

The TaqMan Advanced miRNA cDNA Synthesis Kit (Applied Biosystems, Milan, Italy) was used to synthesize cDNA for miRNA expression analysis. Real-time quantitative PCR (qRT-PCR) was performed for hsa-miR-375-3p on the StepOne real-time PCR system using TaqMan advanced miRNA assays, following the manufacturer’s instructions. miRNA levels were normalized to hsa-miR-24-3p as housekeeping gene. All real-time reactions, including no template controls, were run in triplicate. A pool of cDNA derived from normal adrenal medulla tissue samples was used as the calibrator source. Data were analyzed with the relative quantification (2-ΔΔCt) method, as described elsewhere [42].

### 4.6. qRT-PCR

cDNA for gene expression quantifications was synthesized using a high-capacity cDNA Reverse Transcription Kit (Applied Biosystems, Milan, Italy), following the manufacturer’s instructions.

qRT-PCR was performed in an ABI-PRISM 7900HT Sequence Detector (Applied Biosystems, Milan, Italy) using the relative quantification method (2-ΔΔCt method), as previously described [10]. The genes were analyzed using the following TaqMan assays: *SLC7A5* (Hs00185826_m1); *SLC7A8* (Hs00794796_m1); *PNMT* (Hs00160228_m1); *YAP1* (Hs00371735_m1); *TNKS1* (Hs00186671_m1); *CTNNB1* (Hs00355045_m1); *AXIN2* (Hs00610344_m1); *MCT8* (Hs00185140_m1); and *FZD8* (Hs00259040_s1), all from Applied Biosystems.

Levels of mRNA expression were calculated with the Sequence Detection Software rel. 2.4 (Applied Biosystems, Milan, Italy) using the 2-ΔΔCt method and normalized to the housekeeping gene ACTB (Hs99999903_m1). All real-time reactions, including no template controls, were run in triplicate.

### 4.7. Statistical Analysis

The Kolmogorov–Smirnov test was used to assess the normal distribution of each variable. All data were expressed as means ± standard deviations for variables that were normally distributed and as medians with interquartile ranges (IQR) for those that were not. The Mann–Whitney, Student’s t, and other tests were used, as appropriate, to compare miRNA expression levels in Pheo tissue samples and their normal counterpart. A sub-analysis comparing germinal and sporadic Pheo was also run. Statistical correlations were obtained using rank correlation (Spearman’s rho), when appropriate, for miRNA or mRNA expression levels in Pheo tissues and biochemical or SUV values.

A *p* value < 0.05 was considered statistically significant.

## Figures and Tables

**Figure 1 ijms-23-02413-f001:**
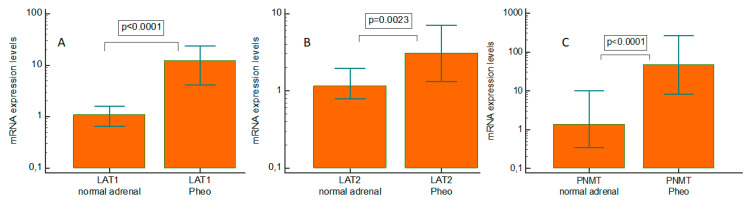
Real-time quantitative PCR gene expression of L-type amino acid transporter 1 (LAT1), L-type amino acid transporter 2 (LAT2), and phenylethanolamine N-methyltransferase in pheochromocytoma (PNMT) specimens and paired normal tissue. (**A**) LAT1; (**B**) LAT2; and (**C**) PNMT. Pheo: pheochromocytoma.

**Figure 2 ijms-23-02413-f002:**
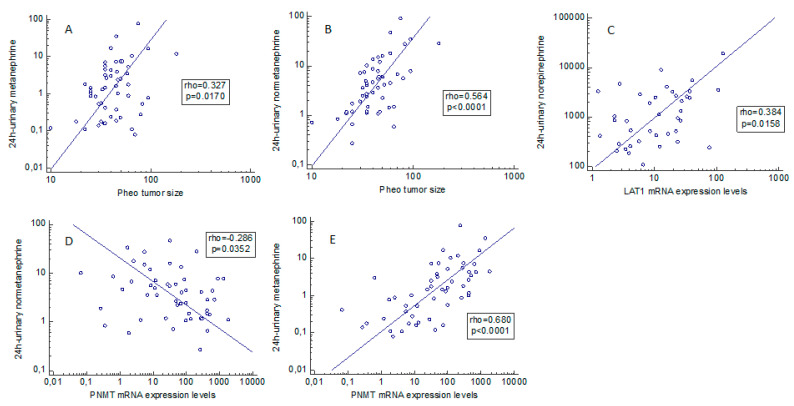
(**A**–**E**) Correlation between pheochromocytoma secretory behavior and clinical/molecular aspects. Statistically significant associations between L-type amino acid transporter 1 and phenylethanolamine N-methyltransferase expression levels and biochemical and clinical data, based on Spearman’s rank correlation. Pheo: pheochromocytoma. PNMT: phenylethanolamine N-methyltransferase.

**Figure 3 ijms-23-02413-f003:**
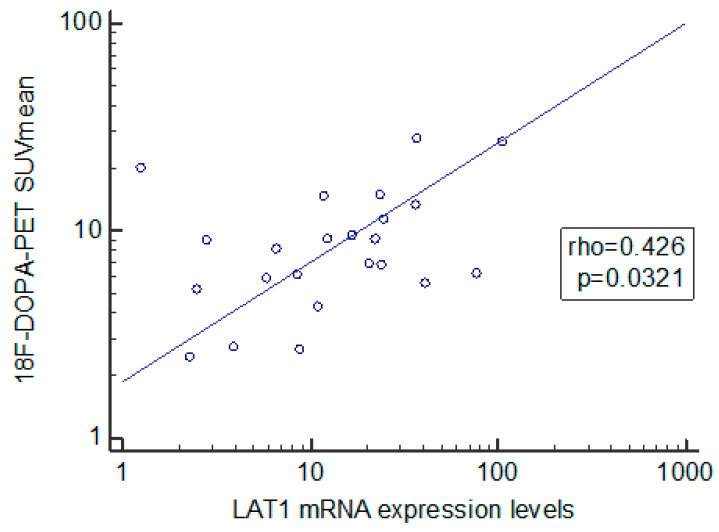
Spearman’s rank correlation coefficient analysis of L-type amino acid transporter 1 and ^18^F-dihydroxyphenylalanine—positron emission tomography.

**Figure 4 ijms-23-02413-f004:**
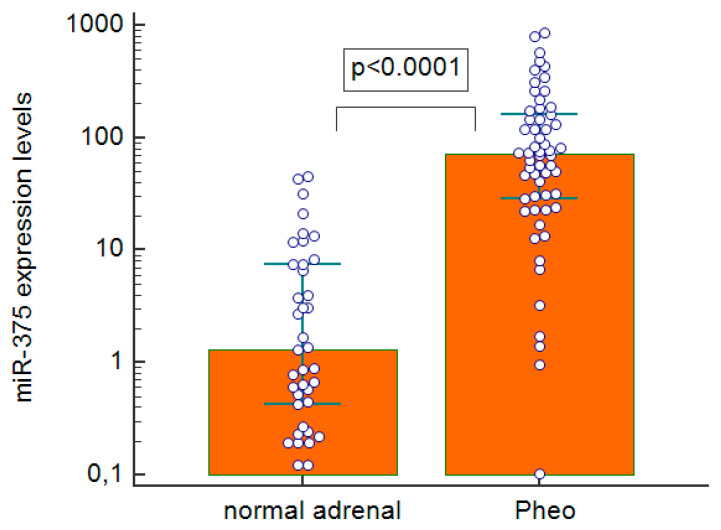
Real-time quantitative PCR expression levels of miR-375 in pheochromocytoma specimens and normal adrenal tissues; Pheo: pheochromocytoma.

**Figure 5 ijms-23-02413-f005:**
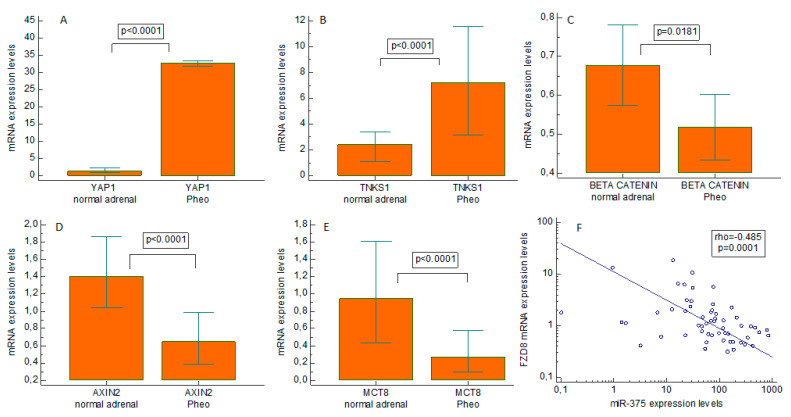
Expression of Hippo and Wingless/Integrated pathways effectors in pheochromocytoma tissues. Real-time quantitative PCR gene expression of Yes-associated protein 1 (YAP1), tankyrase 1 (TNKS1), beta-catenin, axin2, and monocarboxylate transporter 8 (MCT8) in pheochromocytoma specimens and paired normal tissues. (**A**) YAP1; (**B**) TNKS1; (**C**) beta-catenin; (**D**) axin2; and (**E**) MCT8. Negative correlation between Frizzled 8 (FZD8) and the miR-375 expression level (**F**). Pheo: pheochromocytoma.

**Table 1 ijms-23-02413-t001:** Descriptive characteristics of patients with pheochromocytoma.

Descriptive Characteristics of Patients with Pheochromocytoma		
	N = 58	%
**Age at diagnosis (mean ± SD)**	54 years old ± 13	
**Sex**		
M	21	36
F	37	64
**Median follow-up** (median; [IQR])	45 months (23–80)	
**Tumor size** (median; [IQR])	40 mm (32–52)	
**Biological behavior**		
Benign	55	95
Malignant	3	5
**Histological capsular infiltration**		
Present	9	16
**Histological vascular invasion**		
Present	8	14
**Biochemical characterization**		
**24-h urinary epinephrine** (median; [IQR])	119 nmol (35–279)	
**24-h urinary norepinephrine** (median; [IQR])	981 nmol (367–2777)	
**24-h urinary metanephrine** (median; [IQR])	1.46 umol (0.36–4.37)	
**24-h urinary normetanephrine** (median; [IQR])	4.12 umol (1.46–7.78)	
**24-h urinary normetanephrine/metanephrine ratio** (median; [IQR])	2.0 (0.6–19]	
**Genetic background**		
Sporadic	43	74
Familial	15	26
**Positive germline mutation**		
RET	7	12
NF-1	2	3
SDHB	1	2
SDHC	1	2
SDHD	2	3
VHL	1	2
**^18^F-DOPA-PET****SUV max.** (median; [IQR])	13 (8–23) *	12
**^18^F-DOPA-PET SUV mean** (median; [IQR])	8 (6–13) *	3
**Outcome at last follow-up**		
Remission	56	97
Cancer-related death	2	3

* 32/58 patients underwent ^18^F-DOPA-PET at diagnosis.

## Data Availability

Data are available on request due to restrictions, e.g., privacy or ethical restrictions.

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
