# Peer review of "Overexpression of miR-375 and L-type Amino Acid Transporter 1 in Pheochromocytoma and Their Molecular and Functional Implications"

_ijms, 2022, doi:10.3390/ijms23052413_

Round 1

Reviewer 1 Report

In this research manuscript the authors studied  LAT1 and LAT2 expression in pheochromocytoma in 58 patients, and examined the relationship between LATs and the tumor secretory properties, and 18F-DOPA-PET. The manuscript is interesting but I not agree that 58 patients is a large series of patients. I suggest to use the word preliminary results. 

The organisation and structure of the manuscript is weak. The graphs are very disorganized as are the tables. 

Authors should add the PCR and PET/CT imaging protocol images.

The references list should be revised according to the journal instructions.

What are the weaknesses of this manuscript? Publications on the topic already exist and authors should highlight their differences, future guidelines, interest and practical application. These changes should be made in the manuscript.

Reviewer 2 Report

The authors aimed to demonstrate the possible correlations between LATs and 18F-DOP-PET uptake and, as a second aim to evaluate the miR-375 expression in pheochromocytoma. The study evaluated the epigenetic and functional features in this kind of neuroendocrine tumor. The study included samples from 58 adult patients with pheochromocytoma collected during 12 years. The study concluded the LAT1 overexpression that was correlated with 18F-DOPA uptake. Also, the authors proved the mir-375 overexpression, WNt signaling downregulated in pheochromocytoma that would possibly lead to further developments in the Hippo pathway as an oncogenic driver in pheochromocytoma. 

The manuscript needs some improvements:

  • at first, I would improve the title to make it more specific according to the findings of the study;
  • abstract: line 20 - "58 Pheo patients" would be better
  • keywords should not include abbreviated words
  • Table 1: better to include for histological capsular infiltration and histological vascular invasion only the "present" data
  • line 128: Mir-375 should  be MiR-375
  • the titles for Figures 1 and 5 should be improved as there are repetitions
  • Figure 2 - C there is no y-axis explanation
  • Figure 4 - no need for miR-375 again in the x-axis, just normal adrenal and Pheo would be enough
  • Figure 5 - no y-axis name for B
  • in the presentation of the Methods, line 273, it was said that the parents/guardian of the minors signed the informed consent, but all the patients were aged between 25 and 83 years.
  • the authors should verify the reference list to include all the names of the authors
  • some typos should be checked - as is line 209, Authors with a capital letter
  • maybe it would be better if the authors added a paragraph about the importance of these findings for diagnosing or clinical issues in pheochromocytoma.
